# Plug-Based Embolization Techniques of Aortic Side Branches during Standard and Complex Endovascular Aortic Repair

**DOI:** 10.3390/jcm13072084

**Published:** 2024-04-03

**Authors:** Andrea Melloni, Mario D’Oria, Pietro Dioni, Deborah Ongaro, Giovanni Badalamenti, Sandro Lepidi, Stefano Bonardelli, Luca Bertoglio

**Affiliations:** 1Division of Vascular Surgery, Department of Clinical and Experimental Sciences, University and ASST Spedali Civili Hospital of Brescia, 25123 Brescia, Italy; p.dioni001@studenti.unibs.it (P.D.); ongarodeborah@gmail.com (D.O.); stefano.bonardelli@unibs.it (S.B.); luca.bertoglio@unibs.it (L.B.); 2Division of Vascular and Endovascular Surgery, Cardiothoracovascular Department, University Hospital of Trieste ASUGI, 34139 Trieste, Italy; mario.doria88@outlook.com (M.D.); badalamentigiovanni28@gmail.com (G.B.); slepidi@units.it (S.L.)

**Keywords:** endovascular aortic repair, vascular plug, embolization, thoracoabdominal aortic aneurysm, aortic dissection, coils

## Abstract

Vascular plugs are an evolving family of vessel occluders providing a single-device embolization system for large, high-flow arteries. Nitinol mesh plugs and polytetrafluoroethylene membrane plugs are available in different configurations and sizes to occlude arteries from 3 to 20 mm in diameter. Possible applications during complex endovascular aortic procedures are aortic branch embolization to prevent endoleak or to gain an adequate landing zone, directional branch occlusion, and false lumen embolization in aortic dissection. Plugs are delivered through catheters or introducers, and their technical and clinical results are comparable to those of coil embolization. Plugs are more accurate than coils as repositionable devices, less prone to migration, and have fewer blooming artifacts on postoperative computed tomography imaging. Their main drawback is the need for larger delivery systems. This narrative review describes up-to-date techniques and technology for plug embolization in complex aortic repair.

## 1. Introduction

Endovascular repair of complex aortic aneurysms has gained popularity over the last decade due to comparable results with open surgery [1,2], the minor likelihood of complications, their cost-effectiveness [3,4], and their high feasibility [3]. With the rise of endovascular surgery, newer techniques of vessel embolization are being used by vascular specialists to occlude target vessels while treating the main aortic pathology. Arterial embolization can be a planned procedure (e.g., to prevent type 2 endoleak from the inferior mesenteric artery) or a bailout procedure (closure of a directional branch when bridging to the target vessel is not possible).

Historically, vessel embolization was achieved with coils, which are still widely used by interventionists because of their easy applicability and minor cost [4]. Coils should be densely packed in the target vessel to achieve occlusion, and this is often time-consuming and may be less precise than plug use. Some disadvantages of coil embolization could be vessel recanalization over time, non-target embolization, coil migration, and the creation of radiopaque artifacts on follow-up computed tomography angiography (CTA) images [5]. For these reasons, vascular plugs seem to be a valid alternative to coils. The accuracy of the delivery system and the promising results in clot formation corroborate the use of plugs to occlude aortic side branches in many instances. Nevertheless, plugs usually require larger delivery systems as compared to the microcatheter used in coil embolization.

As several devices are present on the market, this narrative review aims to provide a description of commercially available vascular plugs and their possible application in the treatment of complex aortic aneurysms coupled with fenestrated–branched endografts.

## 2. Vascular Plugs: Overview of Available Devices

The description of all the available endovascular plugs on the global market is beyond the scope of this paper. The most frequently employed devices and/or the ones with more robust evidence supporting their use are briefly described below. Techniques and material for plug delivery are standard interventionalist practice, both with 0.035″ platforms and microcatheter ones. In brief, for the larger plugs needing an introducer sheath to be deployed, any guidewire can be advanced beyond the arterial segment, which will be embolized; the introducer (usually 6–7F) is advanced over a regular or stiff guidewire. The guidewire is removed and the plug is advanced towards the target destination, and then the introducer is retracted to allow plug expansion. When angiography confirms the appropriate position of the plug, the occluder is eventually released by unscrewing the delivery wire by turning it counterclockwise. The procedure is the same when using plugs that can be delivered through smaller systems (angiographic catheters or microcatheters), but navigation is usually easier due to their smaller profile.

Their geometric characteristics are summarized in Table 1 and their visual aspect is depicted in Figure 1. As a general principle, instructions for use and published papers suggest a 30 to 50% oversizing to guarantee efficacious vessel occlusion [6]. All the reported plugs are self-expandable devices equipped with proximal and distal markers to enable their precise deployment. Most importantly, these devices can be fully retracted and repositioned multiple times, which allows for safe use as the operator can adjust the position of the device until satisfactory deployment is achieved. Notably, the plug length reported is the unconstrained length, the in situ ultimate length will always exceed this measure, and this drawback must be taken into account in order to prevent unintended collateral occlusion.

### 2.1. Amplatzer Vascular Plugs

The Amplatzer vascular plugs (AVP; St. Jude Medical, St. Paul, MN, USA) I, II, and IV are the most frequently employed AVPs in complex aortic repair. All are deployed by unscrewing a micro-screw on the proximal aspect of the plug. The main differences are configurations and measures, which drive the different applications during complex aortic repair. Moreover, some authors reported shorter occlusion times with AVP II as compared to AVP I [6].

### 2.2. Cera Vascular Plug

The Cera vascular plug (Lifetech Scientific Co., Nanshan District, Shenzhen, China) is a self-expanding, cylindrical, single-disc nitinol plug coated with titanium nitride (TiN) and features three layers of polytetrafluoroethylene (PTFE) membrane. This design is intended to reduce total occlusion time (Figure 1). It is available in 4 to 24 mm diameters, with the largest needing a 9F internal diameter introducer sheath for deployment. 

### 2.3. Micro Vascular Plug

The Micro vascular plug (MVP, Medtronic, Minneapolis, MN, USA) is a cylindrical nitinol frame with the proximal half covered by a PTFE membrane. The MVP metal design resembles a small caval filter. Its mechanism of vessel occlusion is both mechanical (no inflow distally to the plug) and clot-mediated (thrombosis proximal to the graft, extended up to a patent collateral branch). The device is available in four configurations, the largest one being suitable for the occlusion of vessels up to 9 mm in diameter. Notably, the two smaller devices can be delivered through microcatheters, while the two larger plugs can be delivered with 4F and 5F angiographic catheters, respectively.

### 2.4. Azur Vascular Plug

The Azur vascular plug (Terumo Medical Corporation, Somerset, NJ, USA) is available in 5–8–10 mm diameters and its structure has a self-expanding nitinol braided wire frame surrounding a flexible, occlusive membrane (Figure 1). It can be advanced through ProGreat peripheral 2.8F microcatheters (Terumo, Somerset, NJ, USA), available in 140 or 165 cm lengths, in arteries up to 8 mm in diameter.

## 3. Clinical Applications of Vascular Plugs during Endovascular Repair of Complex Aortic Aneurysms

### 3.1. Left Subclavian Artery (LSA)

When thoracic endograft landing is planned to occur in Zone 2 or more proximally, correct management of the LSA becomes crucial, particularly in the case of extensive aortic coverage. Indeed, LSA coverage without revascularization can result in paraplegia, stroke, or left arm ischemia together as a consequence of vertebrobasilar and upper limb arterial insufficiency; on the contrary, a type II endoleak from LSA backflow can maintain perfusion to a thoracic aortic aneurysm or impede adequate false lumen remodeling in the case of Stanford type B aortic dissection treated with thoracic endovascular aortic repair (TEVAR).

Revascularization procedures in patients with LSA coverage during TEVAR are associated with a lower risk of left arm ischemia, stroke, and spinal cord ischemia [7,8]. Even in acute aortic dissection, subclavian/vertebral artery revascularization has been proven to reduce neurological complications [9]. 

LSA revascularization can be performed using different technical modalities [10]:-in an open traditional fashion through carotid to subclavian bypass (CSB) or subclavian to carotid transposition (SCT), with no clear difference between the two in terms of stroke reduction or surgical morbidity [11,12];-in a total endovascular fashion, with parallel graft technique [13], inclusion via physician-modified endografts (PMEG) [14], an in situ laser [15], or needle fenestrations [16].

When surgical revascularization is planned, the LSA needs to be ligated or embolized in its proximal pre-vertebral tract to avoid a Type 2 endoleak. Since it has been reported that multiple coils are often required to successfully embolize the LSA [17,18], vascular plugs are preferred by many operators because only a single device is needed in most cases. On the other hand, plug embolization requires larger delivery systems.

Plug embolization is very much dependent on the diameter and length of the pre-vertebral LSA. The authors previously suggested the use of a large plug, such as the AVP II in lengths exceeding 40 mm. For shorter LSA, the first-generation AVP is suggested if the LSA diameter measures less than 14 mm. In cases of short length and large diameter, the AVP II should be deployed with one disc protruding in the aortic lumen [19]. It is important to know that plugs protruding in the aortic lumen could interfere with TEVAR proximal sealing and cause type IA endoleak, especially if the aneurysm involves the LSA origin. Dissected LSA poses, in fact, some particular technical hurdles that should be addressed, as depicted in Figure 2.

Plug embolization is usually performed after subclavian revascularization and can take place through percutaneous femoral access or upper extremity access. Larger catheters or introducers should be used with caution in narrow upper limb vessels such as the radial artery. On the other hand, plug embolization seems to be less accurate via femoral access because of the longer working distance which compromises the system’s stability [19]. In urgent cases such as aortic ruptures, blunt trauma, or complicated dissections, proximal LSA embolization is performed after TEVAR; therefore, upper extremity access is the only viable alternative [20].

### 3.2. Spinal Segmental Artery Embolization for the Prevention of Spinal Cord Ischemia

Paraplegia and endoleaks are two major issues of extensive aortic endovascular repair in the treatment of complex aortic aneurysms. Paraplegia prevention features both anesthesiologic and surgical skills, and among these, spinal cord preconditioning by the embolization of segmental arteries seems to be a promising tool [21,22]. In minimally invasive segmental artery coil embolization (MISACE), coils are preferably used to embolize segmental arteries before treating patients for aortic conditions. Vascular plugs are used to occlude large segmental arteries that would need a large number of coils, and their use follows the same principles of coil embolization. Plugs are advanced through peripheral percutaneous access, and deployed in the proximal portion of segmental arteries, thus presenting a chance for spinal cord collateral circulation to develop [23].

### 3.3. Occlusion of Aortic Side Branches for Endoleak Prevention

Despite the advances in endovascular aortic repair, endoleaks still represent a challenge in the modern endovascular era. In complex aortic repair, type 2 endoleaks (T2EL) are associated with hampered sac shrinkage over time, lower freedom from reintervention, and complex endoleaks (type 2 + type 1 or 3) [24,25]. The source of persistent sac perfusion is typically in the abdominal segment, often the inferior mesenteric artery (IMA) [26]. Since treatment of T2EL is often suboptimal and reinterventions are frequent [27,28], prevention with pre-emptive plug embolization of aortic side branches could be part of the solution [29]. The embolization of the IMA, as shown in Figure 3, has shown safety and replicability at affordable costs.

While some physicians perform IMA embolization through a brachial access several days before endovascular repair [30], the same procedure could be performed in the same session via femoral access. Some of the same material used for the EVAR procedure would be used in the embolization, thus limiting the cost of the intervention and limiting complications related to access. Samura et al. [31] described their technique for IMA plug embolization which is performed via femoral access at the same time as the endovascular procedure with no additional endovascular equipment except for the AVP IV.

A recent systematic review and meta-analysis including a prospective randomized controlled trial suggested a role of preemptive IMA and/or lumbar artery embolization in standard infrarenal EVAR; sac growth, T2EL, and reintervention were all reduced in the embolization cohort [29,32]. Although the evidence for IMA and lumbar artery embolization in complex aortic repair is unclear, a recent meta-analysis of both prospective and retrospective studies suggests the benefit of occluding any patent IMA and ≥2 mm lumbar arteries to reduce the risk of endoleak in standard infrarenal repair [29], which can lead to overall instability of the endovascular reconstruction. Nevertheless, recent guidelines still do not endorse routine IMA and lumbar artery embolization for T2EL prevention [2] and highlight the higher benefits of IMA rather than lumbar embolization.

When a thoracoabdominal repair is planned, accessory renal arteries can be incorporated with fenestration/branches, if technically feasible, to prevent renal infarction and late renal function decline. When vessel preservation is not possible, accessory renal arteries originating from the dilated segment of the aorta can be embolized to prevent persistent type 2 endoleaks (Figure 4).

### 3.4. Embolization in Aortic Dissection

In the presence of a chronic aortic dissection, some authors have experimented with the use of nitinol plugs in the false lumen (alone or in combination with coils) to promote thrombosis and remodeling [33]. The candy plug technique requires a custom-made device currently only available in Europe in its third generation (William Cook Europe, Bjaeverskov, Denmark). It is a custom-made thoracic stent–graft with three sealing stents with a 26–46 mm diameter range (requiring a 16 to 20F delivery system) and a total length of 97–104 mm, including a distal sleeve segment with a 14 mm wide central fabric channel, which is sutured outside the graft. This technique for false lumen embolization in the distal thoracic segment is the only one with greater scientific evidence to support it (Figure 5). In a multicentric retrospective study of 155 patients treated with a custom-made candy plug device (Cook Medical, Bloomington, IN, USA), 77% achieved early complete false lumen thrombosis, and only 4% experienced false lumen enlargement over time [34]. In the past and still today in certain settings, physician-modified thoracic endografts have been used to serve the same purpose [35,36]. The AVP II has been tried off-label as a false lumen occluder which should be deployed either from the false or true lumen. Compared to the AVP, AVP II has a longer profile and is tri-lobed, which facilitates plug stability at the site of the entry tear. Authors have tried to position it with two disks protruding into the false and one in the true lumen [37].

In very selected cases of acute or chronic type B aortic dissection, entry tear plug embolization was suggested to promote false lumen thrombosis with minimal surgical invasiveness. Such an approach has been described in cases of proximal entry tears in the aortic arch or unfavorable anatomies for standard TEVAR. The AVP II plug was released with one disc in the true lumen and the remaining two in the false lumen, after true-to-false lumen catheterization across the proximal entry tear. Procedural success, however, was scarce, with over 50% of patients requiring early reintervention for persistent entry tear patency [38,39].

### 3.5. Intentional Directional Branch Closure in Complex Aortic Repair

Elective endovascular repair of complex aortic aneurysms and proper thoracoabdominal aneurysms repair with fenestrated–branched endografts (F-B/EVAR) can require tailored treatment with custom-made devices. In case of symptomatic or very large aneurysms, off-the-shelf devices such as the t-Branch (Cook Medical, Bloomington, IN, USA) can be used [40,41]. As the overall feasibility of off-the-shelf endografts is less than 80% for the visceral vessel configuration [3,42], some branches can be intentionally left unused and need to be closed. Other reasons for not bridging a directional branch are chronically occluded vessels, failed catheterization or loss of patency of the target vessel during staged repair, or temporary branch sac perfusion as a spinal cord ischemia prevention strategy [43]. In those scenarios, branches need to be occluded to prevent high-flow continuous type III endoleaks and eventually aortic rupture.

From a technical perspective, alternative methods can be employed to perform plug occlusion of directional branches, detailed as follows.

-As a modification of the original technique from Ferreira et al. [44], Tenorio et al. [45] suggested that branch elongation with a balloon-expandable stent before plug release should be the preferred choice in order to extend the sealing zone; the directional branch should be extended at least 20 mm beyond the branch cuff, and the plug should be released entirely within the stent, with no lobes protruding on the outside (Figure 6a). This configuration would limit any plug migration caused by short landing length inside the branch cuff.

-Alternatively, the “dog bone” technique [46] consists of deploying a balloon-expandable stent-graft inside the branch, sizing it 2 mm larger than the branch itself. It should be inflated to 8 mm, and the proximal and distal portions of the stent should be flared with a larger balloon (4 mm more than the branch diameter). An AVP II should then be released at the beginning of the narrowed part, thus creating a bottleneck effect.-The MVP-7Q (MVP, Medtronic, Minneapolis, MN, USA) has also been used to occlude branches, with promising results in terms of early success and avoiding the use of a further bridging stent–graft, thus reducing overall procedural costs and possibly operating time [47] (Figure 6b). Aside from the economic and technical aspects, the MVP PTFE membrane design guarantees immediate vessel occlusion, which is of paramount importance in symptomatic/ruptured aneurysms, as opposed to the AVP II design which needs time to achieve complete branch thrombosis.

### 3.6. Internal Iliac Artery (IIA)

When extending aortic repair into the iliac arteries, sometimes surgeons must cope with a short distal landing zone on the common iliac artery (CIA) as a consequence of short CIAs, chronic dissection, or aneurysmatic degeneration. This implies the distal sealing zone is located on the external iliac artery (EIA), meaning IIA covering and overstenting (Figure 7a) or inclusion with an iliac branch device (IBD) (Figure 7b) is necessary.

In standard infrarenal aortic repair, plain extension in the EIA without IIA revascularization has shown satisfactory results in terms of endoleak prevention and rates of complications (buttock claudication, erectile dysfunction, colon ischemia, and spinal cord ischemia) with lower costs and complexity than the IBD; long-term results on patency and complications are still pending [48]. Embolization can be performed prior to graft deployment or in the same procedure with no clear differences between the two in terms of ischemic complications [49]. However, the most recent European Society for Vascular Surgery guidelines advise preservation of at least one patent IIA with a Class I recommendation [2].

In the case of non-aneurysmal IIA, the authors suggest performing occlusion with a plug at its origin, thus preserving its distal branches and pelvic collateral circulation. In this setting, plugs seem to be safer than coils in achieving complete occlusion with lower rates of complications as it has been reported that plugs have lower rates of buttock claudication and erectile dysfunction [50,51]. When the sealing zone length is scarce, a short plug such as the AVP should be preferred to a longer plug such as the AVPII, which is chosen in cases of normal length [52]. Proximal positioning of the plug must be avoided in case of aneurysmatic IIA, in which cases the backflow through terminal branches could promote sac enlargement in the IIA. In those cases, a different approach has been described. Small plugs such as the AVP IV or MVP should be released in each main trunk. Considering the large diameter of the IIA and the different anatomies, sometimes, a combination of coils and plugs is necessary to achieve embolization [53].

When IIA preservation is planned as in the case of planned extensive aortic coverage and in young active patients [54], sometimes extending a stent in the distal IIA is not enough to achieve complete sealing. Aneurismal IIA or early bifurcation of the IIA hampers the complete preservation of its side branches [55]. In this case, the authors suggest the preservation of the posterior division of the IIA (superior gluteal artery, SGA) over the pelvic branches due to the higher mid-term patency of this strategy despite similar good short-term results [56] (Figure 7b). Branch abandoning (Figure 7c) is a rare eventuality and is used to avoid aneurysm sac enlargement in event of failure to stent the IIA or SGA after IBD is deployed.

## 4. Discussion

The use of plugs in endovascular aortic surgery is an established alternative to coils for achieving large vessel embolization. A vascular plug, such as those in the AVP family, is a single device made of a nitinol net that is released inside a target vessel before, during, or following the endovascular aortic procedure. Although they are both designed as occlusive tools, coils, and plugs have different delivery systems and eventually different applications.

Plugs provide precise embolization with a single device and can be fully recaptured before final release by readvancing the catheter/sheath over the occluder if their position is unsatisfactory at the test contrast injection. As a disadvantage, they would usually require a larger delivery system, which might be harder to navigate in tortuous anatomies. Coils require microcatheters to be advanced in the target location, while the majority of plugs require a 6 or 7-Fr introducer sheath on a guidewire. Some companies have addressed this issue by producing diagnostic-catheter- or microcatheter-compatible vascular plugs that can be deployed without the help of an introducer sheath. Although coils are preferred in some situations, the use of this new generation of plugs, which have smaller profiles, is feasible on many occasions. In the case of angulated and short arteries or ostial calcifications, plugs might not reach the target location due to delivery system instability and the risk of plug migration. In those scenarios, precision can be increased by inflating a balloon stent–graft over the target vessel once it has been cannulated. This technique, originally proposed by Morikage et al. [31], allows catheter stabilization between the balloon and an introducer sheath while the plug is being released. An evolution of this technique is the so-called “delivery system-F” [57], which consists of a 14 Fr sheath, a 12 Fr long sheath with a side hole, a stiff guidewire as a shaft, and a parallel-inserted delivery catheter navigated through the side hole into the aneurysm sac. The vertical motion and horizontal rotation of the side hole are performed by moving the DrySeal and the 12 Fr sheath, thus securing precision and stability.

The precision of plug delivery is also helped by the fact that plugs can be retracted from a location and repositioned as many times as desired before the final detachment from the delivery wire. When the target location is reached and confirmed at test angiography (Figure 3), unscrewing the delivery system allows for plug release. This is not possible with all available coils, meaning off-target embolization and migration from the original location are rare. Coils that are not sufficiently packed can become occlusive bullets as the blood flow forces their embolization in distal vessels. While the protrusion of coils inside the aortic lumen should be avoided for the risk of peripheral embolization, the protrusion of the first disk of the AVP II is described in stent-graft branches and for LSA embolization, as proposed by the authors. The latter technique was used to embolize the LSA in cases of large and short landing zones but can cause type Ia endoleak by interfering with the TEVAR proximal landing zone, especially if a short distance between the proximal edge of the graft and the protruding plug is present.

Although both procedural and plug embolization times are shorter if compared to coils [58], in high-flow vessels, pure nitinol plugs do not cause vessel thrombosis immediately. This feature of plugs can cause endoleak permanence at short-term follow-up, and therefore it is not suggested by the authors in cases of ruptured or symptomatic aneurysms. Some plugs feature a PTFE membrane inside the nitinol net, intending to immediately cause thrombosis of the vessel. Further studies need to assess this technology together with the durability of the new generation of plugs over time.

Eventually, both nitinol mesh plugs and PTFE-based occluders present significantly fewer artifacts on computed tomography imaging when compared to coils, an effect that is amplified when large vessels (requiring large amounts of coils) are embolized [5]. Given the great importance of precise recognition of endoleaks in the follow-up of complex aortic repair and the rate of reintervention after B/FEVAR [59], when technically feasible, plugs could be used instead of coils to guarantee high-quality CT images to facilitate decision making and proper preoperative planning.

## 5. Conclusions

Plug embolization is a valid adjunctive tool in the endovascular treatment of complex aortic aneurysms. While embolization of side branches is usually achieved in a planned fashion, it may be required under urgent circumstances (for instance, when using off-the-shelf branched devices for treating symptomatic or ruptured complex aortic aneurysms in patients with a solitary kidney); for this reason, these devices should be available and in stock at centers that perform these procedures, and surgeons should be familiar with their use. Plugs can adapt to different anatomies and have different uses while sharing some common functioning principles. Despite shorter procedural and occlusion times and fewer recanalization rates over time, they are not applicable in all situations, and therefore coils are still used on some occasions. More evidence should be gathered on preemptive embolization techniques, with a particular focus on indication (e.g., diameter thresholds) and clinical impact, since evidence on plug embolization in aortic procedures is mostly extrapolated from studies regarding infrarenal EVAR.

## Figures and Tables

**Figure 1 jcm-13-02084-f001:**
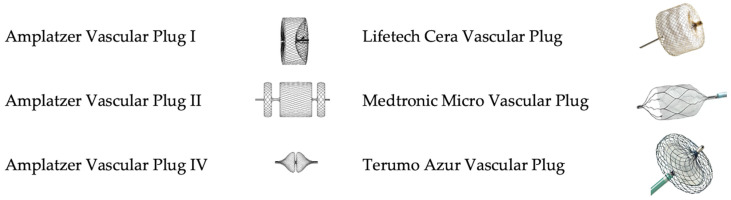
Graphical representation of the vascular plugs described in the present review. AVP I is a single disc with a 4 to 16 mm diameter; it is designed to embolize a vessel with a short neck/landing zone. AVP II is composed of three discs of multi-layered nitinol, available from 3 to 22 mm in diameter. It is longer than the AVP I and designed to promote rapid occlusion where a long sealing zone is available. For both AVP I and II, the delivery wire length is 135 cm. AVP IV is a double-disc plug provided in 4 to 8 mm diameter. It is delivered via a 155 cm long nitinol wire and is designed for embolization of small vessels in tortuous anatomies. The Cera, the Micro vascular plug, and the Azur vascular plug combine a metallic frame with a membrane to promote faster vessel occlusion.

**Figure 2 jcm-13-02084-f002:**
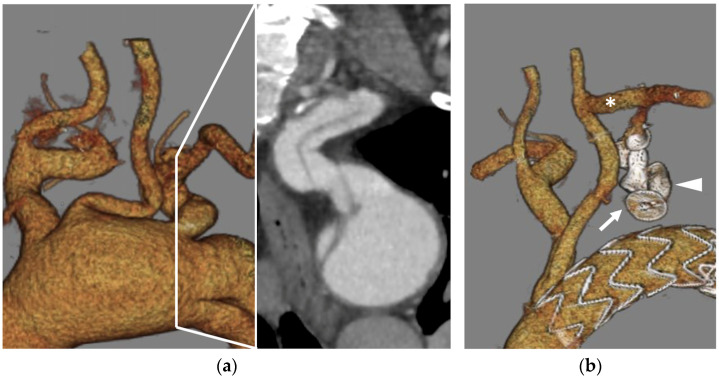
False and true lumen left subclavian artery (LSA) embolization in the treatment of post-dissection aortic arch and thoracoabdominal aneurysm: (**a**) preoperative CTA 3D volume rendering of a post-dissection aortic arch aneurysm involving the LSA with a lens showing a parasagittal detail of the dissected LSA; (**b**): postoperative imaging showing a frozen elephant trunk hybrid graft, a 16 mm Amplatzer vascular plug (AVP) 2 in the true lumen (arrow) and a 12 mm AVP I in the false lumen (arrowhead) with a patent carotid-subclavian bypass (*). Both plugs were deployed through a 6F Cook Flexor introducer sheath via a percutaneous trans-radial approach and alternate catheterization of the true and false lumens through a distal re-entry tear located at the level of the internal mammary artery origin.

**Figure 3 jcm-13-02084-f003:**
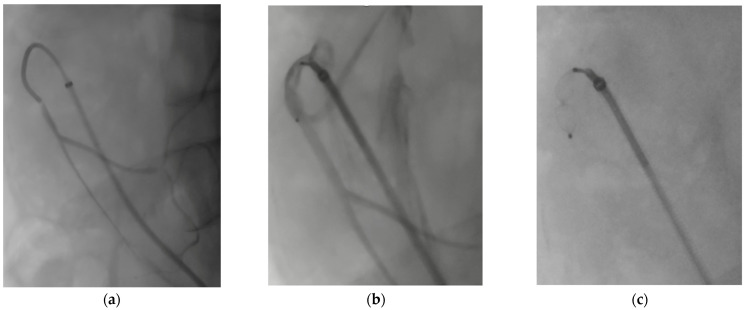
Pre-emptive inferior mesenteric artery (IMA) embolization to prevent Type 2 endoleak: (**a**) intraoperative angiographic images showing selective catheterization from transfemoral access; (**b**) a 6 mm Amplatzer vascular plug (AVP) IV is advanced at the origin of the patent IMA via a 5F Berenstein diagnostic catheter (Tempo Aqua, Cordis, Santa Clara, CA, USA); angiography from the catheter confirms the position proximal to a patent collateral branch; (**c**) controlled release of the plug in the intended position.

**Figure 4 jcm-13-02084-f004:**
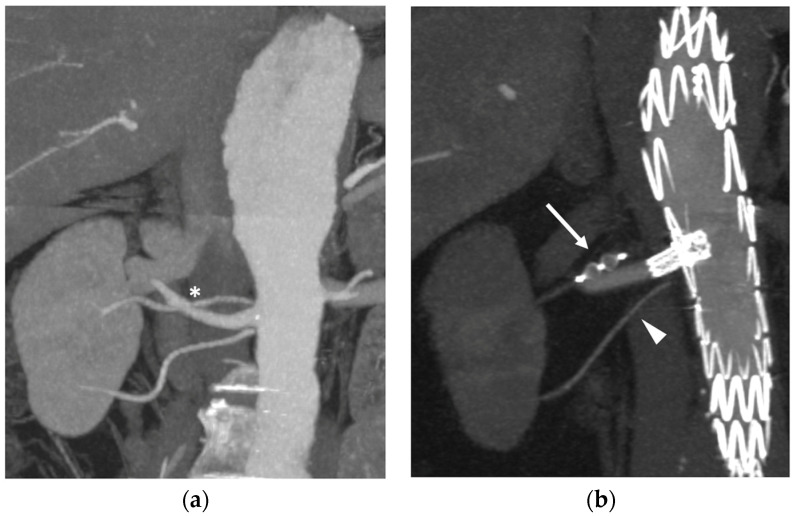
Accessory renal artery embolization during a fenestrated thoracoabdominal aortic repair: (**a**) CTA in a para-coronal plane showing three patent right renal arteries can be seen on preoperative imaging. An asterisk (*) indicates the one arising from the aneurysm; (**b**) Intraoperative angiography demonstrating the superior polar renal artery embolization with an Amplatzer vascular plug IV (arrow). The middle one is bridged to the dedicated fenestration, and the inferior (arrowhead) is covered by the aortic graft in a non-aneurysmal portion of the aorta.

**Figure 5 jcm-13-02084-f005:**
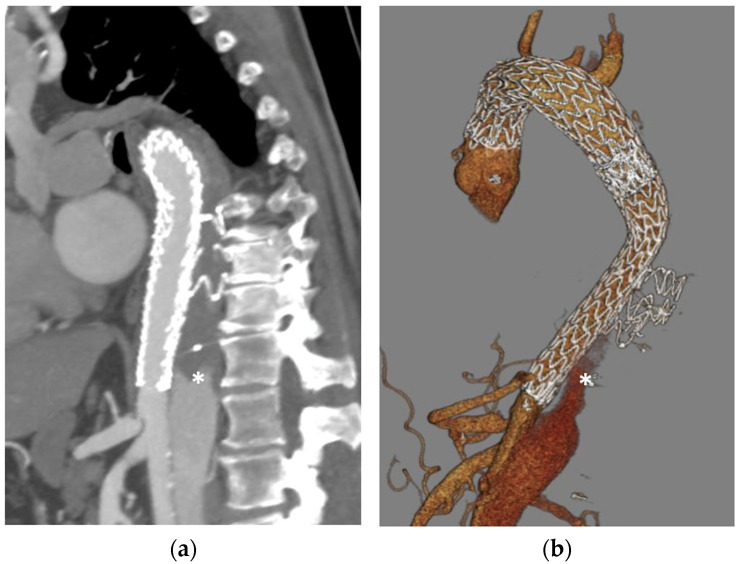
Aortic false lumen embolization with a third-generation custom-made candy plug (Cook Medical, Bloomington, IN, USA) in a chronic dissection with aneurysmal evolution treated by frozen elephant trunk, true lumen thoracic EVAR (W.L. Gore, Flagstaff, AZ, USA), and false lumen embolization: (**a**) 3D MPR image and (**b**) a parasagittal view of the CTA 3D volume rendering showing false lumen thrombosis achieved in the cranial portion of the descending thoracic aorta with persistent retrograde perfusion below the occluder (*).

**Figure 6 jcm-13-02084-f006:**
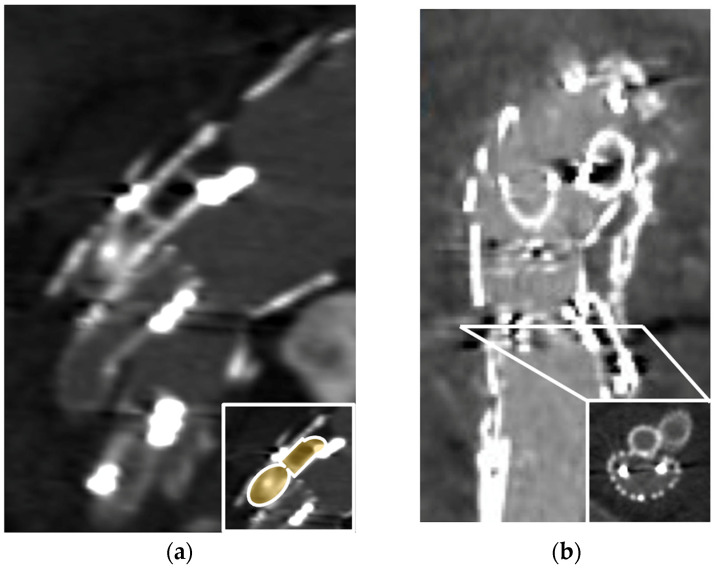
Branch embolization using vascular plugs: (**a**) postoperative CTA para-sagittal view showing celiac trunk branch (CT) embolization with a 12 mm Amplatzer vascular plug II (the silhouette is highlighted in the small box) in a patient treated with a t-Branch device and a chronic CT occlusion. The plug is released after branch extension with a 9 × 37 balloon-expandable stent-graft; (**b**) postoperative CTA para-coronal view embolization of directional branches for both renal arteries with a Micro vascular plug without pre-emptive branch covered-stenting; the small box indicates a cross-sectional view perpendicular to the coronal main image. Notably, no blooming artifacts are visible because of the plug’s presence. The MVP-7Q is suggested for 6 mm branches and the MVP-9Q for 8 mm branches. The occluder is advanced to the target location through 4F or 5F diagnostic catheters.

**Figure 7 jcm-13-02084-f007:**
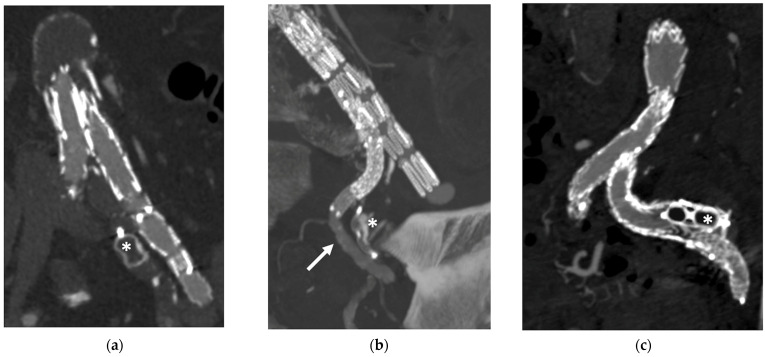
Plug embolization in the internal iliac artery (IIA) territory as visible on postoperative CTA imaging. (**a**): an Amplatzer vascular plug (AVP, *) I is employed to occlude the main trunk of the IIA before overstenting it; (**b**): the IBD should be elongated with a self-expandable stent in the SGA and pelvic branches should be occluded using small plugs such as the AVP IV, MVP, or even coils. In this patient, an AVP IV occludes the anterior branch of the IIA (*) and the bridging stent is deployed into the superior gluteal artery (arrow); (**c**): Sometimes, after IBD deployment, catheterization or bridging failure of the SGA is encountered, with backflow from the hypogastric branch causing a type III endoleak. In this case, the authors suggest distal branch embolization with coils or microplugs and subsequent plug occlusion of the branch; here, an AVP II (*) is deployed in the hypogastric branch after failed bridging with a bridging stent.

**Table 1 jcm-13-02084-t001:** Sizing and delivery system compatibility for the vascular plugs mentioned in the present review, according to the instructions for use of the single device.

Plug Name	Plug Diameter (mm)	Plug Length (mm)	Delivery System Minimum Internal Diameter (Inches)
AVP I *	4–6–8–10–12–14–16	7–8	0.056–0.088
AVP II *	3–4–6–8–10–12–14–16–18–20–22	6–18	0.056–0.098
AVP IV *	4–5–6–7–8	10–13.5	0.038 (diagnostic catheters)
Cera vascular plug	4–24	7–14	0.055–0.122
MVP **	5.3–6.5–9.2–13	12–12–16–18	0.021 microcatheter–0.043
Azur vascular plug	5–8–10	13.5–18.5	0.027 PG pro peripheral microcatheter

* AVP: Amplatzer vascular plug, ** MVP Micro vascular plug.

## Data Availability

No new data were created or analyzed in this study. Data sharing does not apply to this article.

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
