# Peer review of "Plug-Based Embolization Techniques of Aortic Side Branches during Standard and Complex Endovascular Aortic Repair"

_jcm, 2024, doi:10.3390/jcm13072084_

Round 1

Reviewer 1 Report

Comments and Suggestions for Authors

Thank you for the chance to review the manuscript by Melloni et al.  A brief narrative review of the most frequetnly applied vascular plugs is given. The manuscript provides some necessary and interestin information for the vascular surgeon to be. 

I have a few comments that might need to be adressed before publication of the paper. 

General:

- I am not a native speaker myself, but at certain sentences some either definite or indefinite articles appear to be missing. Please revise carefully. 

- 7 authors seem quite a lot for a narrative review. Please consider revision according to the journal's instructions. 

Abstract: please specify whether this is a narrative or systematice review. 

Introduction: 

- "as long as its cost-effectiveness [3,4] and high feasibility" doesn't sound like a good english sentence to me. Please revise. 

- please provide some examples of what target vessel might need to be occluded during what procedure. For the unfamiliar reader this could be valuable information. 

- please also introduce disadvantages of pluugs over coils, such as the need for larger devices to plasce plugs compared to a coil microcatheter. 

section 2:

- please provide the kind of sheath needed for introcudtion along with vascular plug size in table 1; 

- is the oversizing of 30-50% based on the reference 6 only? are these IFU instructions? what is the authors personal experience with oversizing? too much oversizing might result in a plug too long to fit i.e. a short subclavian artery harboring the risk of occlusion of the vertebral artery. Please revise and elaborate for readers less familiar with the technique. 

- what are the re-deploy fixations? all screws?

- have you considered introducing the medtronic talent occluder along with the other plugs? this is one of the large diameter workhorses for many years on the european continent. 

section 3:

-Fig. 6A could be improved in visbility. 

- colic ischemia? do you mean colon ischemia?

Conclusion:

- facile? do you mean familiar?

Comments on the Quality of English Language

please see above

Author Response

Reviewers response form

Reviewer 1

Thank you for the chance to review the manuscript by Melloni et al.  A brief narrative review of the most frequetnly applied vascular plugs is given. The manuscript provides some necessary and interestin information for the vascular surgeon to be. 

I have a few comments that might need to be adressed before publication of the paper. 

General:

- I am not a native speaker myself, but at certain sentences some either definite or indefinite articles appear to be missing. Please revise carefully.
A revision was performed also with the use of a dedicated software and corrections were made.

- 7 authors seem quite a lot for a narrative review. Please consider revision according to the journal's instructions. 
We couldn’t find a maximum number of authors in the instructions available at https://www.mdpi.com/journal/jcm/instructions. All authors contributed to the final version of the proposed manuscript as described in the dedicated section. We leave the decision to the Editors if any authors should be removed.

Abstract: please specify whether this is a narrative or systematice review.
The wording was changed accordingly.

Introduction: 

- "as long as its cost-effectiveness [3,4] and high feasibility" doesn't sound like a good english sentence to me. Please revise.
The wording was changed accordingly.

- please provide some examples of what target vessel might need to be occluded during what procedure. For the unfamiliar reader this could be valuable information.
A brief note was added, to introduce the topic without repeating the same information included in the body of text.

- please also introduce disadvantages of pluugs over coils, such as the need for larger devices to plasce plugs compared to a coil microcatheter.
A brief note was added, however the introduction section was not meant to be a discussion section, an extensive between coils and plugs is given throughout the article.

section 2:

- please provide the kind of sheath needed for introcudtion along with vascular plug size in table 1;
No specific kind of sheath is required as long as it of the specified internal diameter and shorter than the delivery system mentioned in the table legend.

- is the oversizing of 30-50% based on the reference 6 only? are these IFU instructions? what is the authors personal experience with oversizing? too much oversizing might result in a plug too long to fit i.e. a short subclavian artery harboring the risk of occlusion of the vertebral artery. Please revise and elaborate for readers less familiar with the technique. 
The required corrections were made. The authors suggestion for the management of LSA plug embolization are reported in the dedicated section and should not be duplicated here.

- what are the re-deploy fixations? all screws?
Yes, they are.

- have you considered introducing the medtronic talent occluder along with the other plugs? this is one of the large diameter workhorses for many years on the european continent.
The old candy plug as the more recently described off-label solutions with the Gore TAG device were not mentioned as the use of dedicated devices is in the Authors’ opinion is preferrable, the existence of such techniques was added and referenced.

section 3:

-Fig. 6A could be improved in visbility. 
Unfortunately the quality of the original CTA is suboptimal so the windowing aims at showing both the plug and the branch.

- colic ischemia? do you mean colon ischemia?
The typo was corrected.

Conclusion:

- facile? do you mean familiar?
The typo was corrected.

Reviewer 2 Report

Comments and Suggestions for Authors

Journal: JCM (ISSN 2077-0383)

Manuscript ID: jcm-2909944

Type: Review

Title: Plug-based embolization techniques of aortic side branches during standard and complex endovascular aortic repairRevision suggestions for the article are listed below.

1-Regarding the 'Abstract' section

1-The terms specified as abbreviations should first be written in their complete form.

2—In the article titled 'Plug-based embolization techniques', first of all, a sentence explaining the relevant term and an explanation of the techniques to be included in the article should be made. A sentence explaining the superiority of these techniques with objective data regarding their use in 'aortic side branches' compared to other techniques should also be included. In this respect, the 'Abstract' section should be restructured.

2. Regarding the 'Introduction' section

1-The terms specified as abbreviations should first be written in their full form.

2-In this section, sentences explaining the primary and secondary purposes of the study should be included before including subheadings.

3. Regarding the 'Vascular plugs: overview of available devices.'

1-If the 'plug names' and 'methods' in Table 1 are taken from a study, the study reference must be stated.

2-The subheadings under this heading have been numbered incorrectly.

3- The abbreviations under the heading 'Cera vascular plug' should first be written in their full form.

4. Regarding the 'Clinical applications of vascular plugs during endovascular repair of complex aortic aneurysms'

1-Abbreviations should be included in their full form first.

3. Regarding the 'Discussion' section

1-For this section, objective literature data should be compared with other existing methods for the same process based on the method given in the 'article title.' It should be expanded and supported by studies proving the superiority of the relevant method.

5. Shortcomings throughout the article

1-For all figures included in the article, information about the imaging method used and which segment of the artery is shown anatomically is missing. In this respect, an addition is recommended. If the figure is quoted from a source, it is recommended that a reference to it be made and copyright information provided.

2-Explanation in terms of techniques and equipment regarding the procedures applied in the text of the article is insufficient, so explanatory sentences should be included (for example, EVAR procedure, TEVAR procedure, etc.).

5-There are spelling and spelling errors in the article. It should be reconsidered from this perspective.

6-Explanatory sentences should be included for the different revascularization methods included in the article.

Author Response

Reviewer 2

Title: Plug-based embolization techniques of aortic side branches during standard and complex endovascular aortic repair Revision suggestions for the article are listed below.

1-Regarding the 'Abstract' section

1-The terms specified as abbreviations should first be written in their complete form.
The wording was changed accordingly.

2—In the article titled 'Plug-based embolization techniques', first of all, a sentence explaining the relevant term and an explanation of the techniques to be included in the article should be made. A sentence explaining the superiority of these techniques with objective data regarding their use in 'aortic side branches' compared to other techniques should also be included. In this respect, the 'Abstract' section should be restructured.
The structure of the abstract was changed accordingly.

  1. Regarding the 'Introduction' section

1-The terms specified as abbreviations should first be written in their full form.
The sentence was changed accordingly.

2-In this section, sentences explaining the primary and secondary purposes of the study should be included before including subheadings.
The study is a narrative review so it cannot be structured in primary and secondary endpoints/outcomes/purposes. The aim of the paper is usually the last paragraph of the introduction section and so it is in the present paper.

  1. Regarding the 'Vascular plugs: overview of available devices.'

1-If the 'plug names' and 'methods' in Table 1 are taken from a study, the study reference must be stated.
Table 1 is original.

2-The subheadings under this heading have been numbered incorrectly.
The subheadings were renumbered.

3- The abbreviations under the heading 'Cera vascular plug' should first be written in their full form.
The wording was changed accordingly.

  1. Regarding the 'Clinical applications of vascular plugs during endovascular repair of complex aortic aneurysms'

1-Abbreviations should be included in their full form first.
The wording was changed accordingly.

  1. Regarding the 'Discussion' section

1-For this section, objective literature data should be compared with other existing methods for the same process based on the method given in the 'article title.' It should be expanded and supported by studies proving the superiority of the relevant method.
The entire article is supported by 62 references from the literature. The comparison is conducted in the section 3 for each category/subsection, for instance a meta-analysis is cited for the comparison between plug and coil embolization in the IIA territory (ref 50 Wong et al.). The authors chose to discuss the single application in the third chapter, with images from real cases the author operated on and further technical explanation in figure legends.

  1. Shortcomings throughout the article

1-For all figures included in the article, information about the imaging method used and which segment of the artery is shown anatomically is missing. In this respect, an addition is recommended. If the figure is quoted from a source, it is recommended that a reference to it be made and copyright information provided.
All images are original. Integration of the required information, when missing, was added.

2-Explanation in terms of techniques and equipment regarding the procedures applied in the text of the article is insufficient, so explanatory sentences should be included (for example, EVAR procedure, TEVAR procedure, etc.).
The paper is an invited contribution regarding “Advance in Clinical Application of Embolization Techniques”, so the authors’ perspective is that the Reader is already familiar with basic notions regarding the endovascular aortic treatment and can benefit from a more detailed overview of a specific topic rather than a generic description of standard procedures. More details are provided in the single figures to avoid repetition. A general paragraph was added in section 2.

5-There are spelling and spelling errors in the article. It should be reconsidered from this perspective.
A further grammar check by two authors and a dedicated software was carried out to check typos.

6-Explanatory sentences should be included for the different revascularization methods included in the article.
The article is about vessel embolization rather than revascularization, the modality of surgical revascularization for instance of the LSA is far beyond the aim of the paper. All the mentioned vessels, when not occluded, are preserved by fenestration or directional branch mating with dedicated bridging stents, but the description of the fenestrated/branched devices are not the purpose o

Round 2

Reviewer 2 Report

Comments and Suggestions for Authors

Dear Author,

In the article, radiological images of different patients 'reportedly performed by your team' were used. Please make an explanation about the official procedures (patient consent) for the use of these data in the 'Acknowledgments' section. The changes made in light of the proposed revisions were found sufficient.